# Farmscape Composition and Livelihood Sustainability in Deforested Landscapes of Colombian Amazonia

**Lisset Pérez Marulanda** [1,2,*], **Patrick Lavelle** [3], **Martin Rudbeck Jepsen** [2],
**Augusto Castro-Nunez** [1], **Wendy Francesconi** [1], **Karen Camilo** [1,4], **Martha Vanegas-Cubillos** [1],
**Miguel Antonio Romero** [1], **Juan Carlos Suárez** [5], **Antonio Solarte** [6] and **Marcela Quintero** [1]

1   International Center for Tropical Agriculture (CIAT), km 17 recta Cali-Palmira, Cali 763537, Colombia;
    augusto.castro@cgiar.org (A.C.-N.); w.francesconi@cgiar.org (W.F.); K.Camilo@cgiar.org (K.C.);
    m.vanegas@cgiar.org (M.V.-C.); m.a.romero@cgiar.org (M.A.R.); m.quintero@cgiar.org (M.Q.)
2   Department of Geosciences and Natural Resource Management, University of Copenhagen,
    DK-1350 Copenhagen, Denmark; mrj@ign.ku.dk
3   Institute of Ecology and Environmental Sciences (Biodis), Paris Sorbonne University, 75005 Paris, France;
    patrick.lavelle@ird.fr
4   Department of Economics, Universidad ICESI, Cali 760031, Colombia
5   Facultad de Ingeniería, Universidad de la Amazonía, Florencia, Caquetá 184010, Colombia;
    ju.suarez@udla.edu.co
6   Centro para la Investigación en Sistemas Sostenibles para la Producción Agropecuaria (CIPAV),
    Cali 760042, Colombia; antonio@fun.cipav.org.co
*   Correspondence: lisset.perez@cgiar.org; Tel.: +57-(2)-445-0100; Fax: +57-(2)-445-0073

**Abstract:** In this article, we operationalized a sustainability framing based on the Sustainable Rural Livelihood Resources Framework (SLF), which consists of five capitals—human, physical, social, financial, and natural. We proposed a sustainability index (SI) for two landscapes dominated by two agricultural systems: cattle ranching and small-scale family agriculture. Farm variables within each capital were analyzed using confirmatory factor analysis. Key variables were identified and index values were calculated for each capital. These were combined through a set of simultaneous equations to estimate farm-specific capitals and SI from the observed farm variables. Principal component and cluster analyses were used to group the farms according to their index scores and to further compare their characteristics. Furthermore, with the purpose of comparing the index scoring with an independent metric, a landscape indicator, which comes from a continuous forest, was calculated. From the results, the capitals that contributed to a higher SI score the most were financial and physical. As cattle ranching was associated with higher economic returns and infrastructure investments, this livelihood was identified as the most sustainable. Yet, cattle ranching has been a deforestation driver in the region. These results are attributed to the current conceptual framework design, which gives greater weight to material and economic variables; therefore, it generates a weak sustainability measure. Although the framework allowed us to identify land-use alternatives that could improve SI scores (i.e., silvopastoral systems), corrections to the proposed framework and methodological approach will need to include additional environmental benefits currently unaccounted for. Farmers that use their farms for conservation purposes should be recognized and compensated. An improved environmentally focused SI operational framework could help to endorse and promote sustainable livelihoods and to generate a strong sustainability measure.

**Keywords:** silvopastoral system; confirmatory factor analysis; sustainable land-use; farmscape; sustainability

## 1. Introduction

Sustainability is a wide and multidimensional concept that should be measured using variables selected for their conceptual meaning to provide relevant and reliable indicators [1]. Several indicators, metrics, and indices have been proposed for assessing the sustainability of smallholder farms [2]. However, their application under different socioeconomic and geographic contexts does not guarantee generating reliable results [1]. It is therefore essential to identify site-specific and suitable indicators to evaluate sustainability and generate objective information to guide policymakers along a strong sustainable development path. Additionally, although measurement efforts of several national and international institutions have been made, sustainability accounting has weaknesses and requires a more highly developed theoretical framework, rather than only statistical methods, and the use of relevant criteria for generating unbiased information [1].

The expansion of the agricultural frontier, especially for cattle ranching, is the primary cause of deforestation and forest degradation in Colombia [3]. Approximately 47% of forest loss in the Colombian Amazon is concentrated in Caquetá, which is at the Amazon frontier and where agriculture and cattle ranching increased by almost 40% from 1999 to 2017 [4]. Cattle ranching generates 6% of the employment and 1.6% of the national gross domestic product [5]. Among the deforestation causes in Caquetá are the armed conflict, especially the establishment and eradication of illicit crops; the incentivizing deforestation along building roads for the oil industry, extractive mining, and cattle ranching [6]. However, cattle ranching is the predominant economic activity responsible for about 35% of forest loss and/or degradation [7,8]. In 2019, 10,000 hectares of forest were converted to livestock production in this department [3]. Consequently, as part of the Paris agreement commitment [9], the Colombian government pledged to implement more sustainable alternatives to traditional cattle ranching [10]. In addition, there are several efforts to incentivize sustainable practices as reforestation and conservation [11]. However, determining improvements in the sustainability of agricultural systems in Colombia, and particularly in the department of Caquetá, is required.

As a methodological approach for measuring sustainability, we use the Sustainable Rural Livelihood Resources Framework (SLF) [12], which contains five dimensions or capitals (natural, physical, human, financial, and social). These, in turn, are composed of a series of indices calculated from farm variables. According to the SLF, sustainability can be achieved through access to a variety of livelihood resources, which are contained in the five capitals. This implies that natural systems should be allowed to recover following stress, and none of the capital should be constantly dominant over the others [12,13]. However, the material dimension has double accounting in the physical and financial capitals. The SLF has been widely used in programs related to poverty alleviation as a basis for monitoring and evaluating these initiatives [14,15].

The SLF has been used in the analysis of complex rural livelihoods and in research on natural resource management [15–19]. This framework allows interpreting links among contexts, strategies, and outcomes, which are missing in other sustainability approaches. This approach allows farmers and decision makers to assess changes in environmental and economic conditions and to identify factors that may have led to particular outcomes in regard to livelihood strategies [2]. Hence, the framework can examine the benefits and impact of different interventions on households and communities [14,15]. For these reasons, the SLF approach has mostly been used for assessments of local-level and assistance programs [15,20] and for turning specific priorities into actions [16]. Despite the documented strengths of this conceptual framework, we evaluated its weaknesses in its application to the particular case of the Colombian Amazon.

Given the need to mitigate deforestation activities in the Amazon-bordering department of Caquetá in Colombia, and to assess the sustainability of agricultural livelihood practices in the region, we adapted and explored the applicability of the SLF methodological approach. Our objective was to evaluate a standardized measurement of sustainability among small-scale agricultural and dairy production activities, which are the two predominant farming systems in four municipalities of Caquetá: Morelia, Albania, Belén de los Andaquíes, and San José de Fragua (Figure 1). In this region,

two different land-use systems dominate: cattle ranching for milk production, and a system dominated by smallholder farmers who grow subsistence crops such as plantain and cassava. Cattle ranching prevails in the rolling hills, while small-scale family agriculture takes place on the hillsides with steeper slopes.

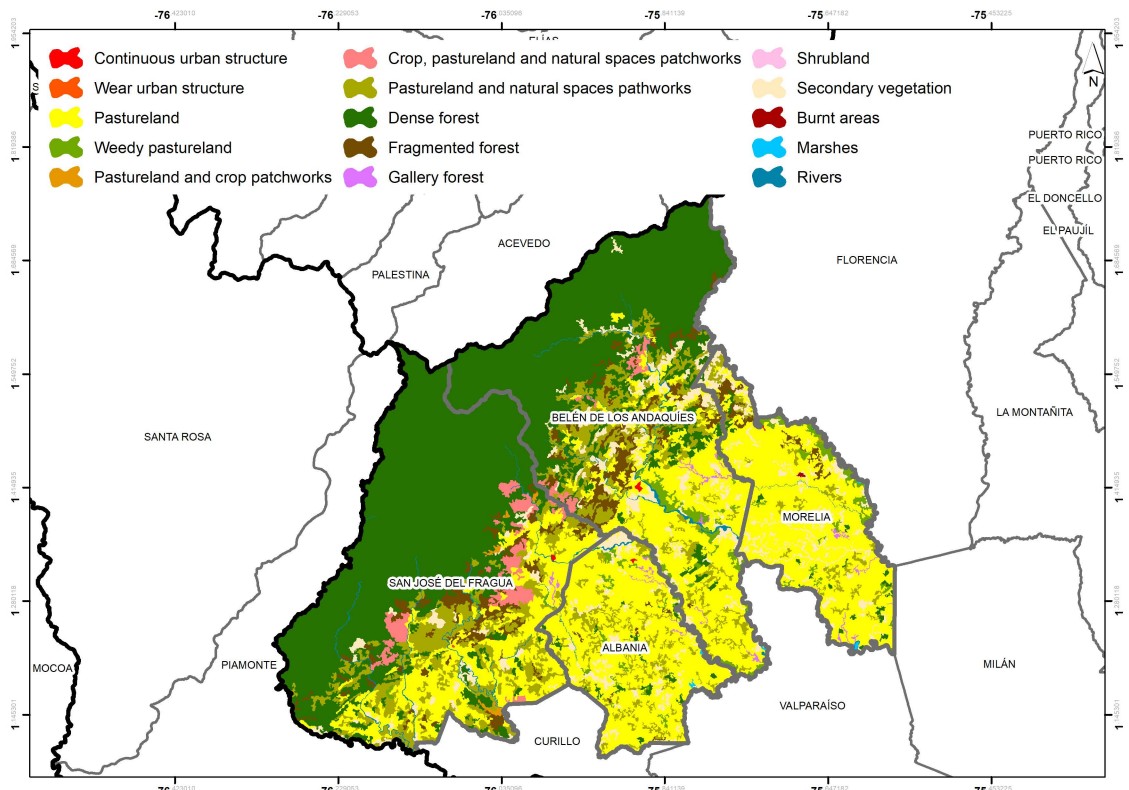

**Figure 1.** The study site. The map represents the four municipalities of Caquetá included in the study area. Two well-differentiated types of landscapes are observed. Areas colored in yellow represent the foothills with pastureland as the principal land cover, where cattle ranching is predominant. The dark green area shows that the predominant land cover is a dense forest in the mountains, where small-scale family agriculture is the principal productive activity. In the center, at the junction of the two landscapes, a transition area is observed, where shrub-land and fragmented forest dominate. Credit: Fabio Alexander Castro.

Also, we wanted to determine if the conceptual framework is suitable for measuring sustainability in the Colombian Amazon by comparing its application with an independent metric of the group of capitals, which is a landscape indicator. It is a terminal indicator of change because the landscapes come from a continuous forest. Finally, we suggested which alternative land-use systems could increase sustainability at the farm level and understood how the land-use composition of these systems within the farm area (farmscape) affects livelihood sustainability. Moreover, we proposed that the type of home that is more specialized in livestock production needs a more diverse landscape and more food security, while for those who are only in small-scale agriculture, it is necessary to add a livestock component to make organic fertilizer and generate sources of income.

## 2. Materials and Methods

### 2.1. Household Data

Interviews were conducted with 341 households, 176 (51.6%) in the Andean foothills (with cattle ranching as the predominant activity) and 165 (48.4%) in the hillside area, where small-scale family agriculture is predominant. According to the Colombian census carried out in 2014 [21], the sample is

representative of the 22% of the farmers who are mainly dedicated to cattle ranching in the foothills and of the 15% who practice small-scale agriculture in the mountains. We used a stratified optimal random sampling strategy across the number of rural households in the municipalities to account for the number of farms (95% confidence). Stratified optimal random sampling is convenient when within-group variability varies widely across groups; it was suitable, since we had two well-differentiated groups in the two landscapes, this allowed us to address productivity variation due to landscape characteristics, which could affect the sustainability. The information was collected from March to October 2016.

The household-level survey's primary purpose was to document the current socioeconomic conditions and farming practices engaged by the farmers. Later, this information was further synthesized to estimate the livelihood sustainability index (for further details, see Supplementary Material Table S1 and Figure S1).

Eight local survey takers were trained to apply the questionnaire and follow the data collection procedure. We interviewed the heads of households or their spouses, decision-makers, and the people most familiar with household duties and productive activities. The interviewed households had to fill out a consent protocol. The survey data were collected using Android tablet devices with CSPro Software 6.2 and 6.3 [22].

### 2.2. Data Treatment

Following the survey application, the data were checked to eliminate entry and measurement errors. A quality control procedure was followed, which included an exploratory analysis for highly correlated variables (Spearman higher than 0.7) and removal of variables as indicators when more than 20% of the values were missing. In addition, we applied a statistical diagnosis (distribution analysis, descriptive statistics) and data outliers were removed. Before determining the final database, several composite variables and indices proposed by the literature were calculated. These included Household Dietary Diversity Score (HDDS) [23], dependency rate, and schooling rate. Last, the final database was composed of 22 variables for cattle ranching and 24 variables for small-scale family agriculture, which were analyzed using confirmatory factor analysis (CFA) estimations.

### 2.3. Confirmatory Factor Analysis

The CFA method estimates the capital indices (latent variables) from normalized farm variables [24,25]. The weighting of the variables harnesses the multivariate nature of the CFA to obtain optimal weights from the used data. There are pros and cons of the weighting based on statistical methods. The main advantage is that good mathematical properties decide the set of weights which explains the largest variation in the observed farm variables, while the main disadvantage is small weights are given to variables which have little variation [26,27]. The sustainability index (SI) is derived from the five capitals of the adapted model. Natural capital includes natural resource stocks (soil, water, forest, and others) and environmental services (water, forest, and high air quality). Financial capital comprises savings, income, and credit. Human capital incorporates skills, knowledge, ability to labor, and good health. Social capital involves social relations, associations, and the capacity to make decisions. Finally, physical capital is represented by infrastructure and production equipment, tools, and technologies [12,15]. The graphical display of our complete model is depicted in Figure 2, and the equations system is described in Equation (1).

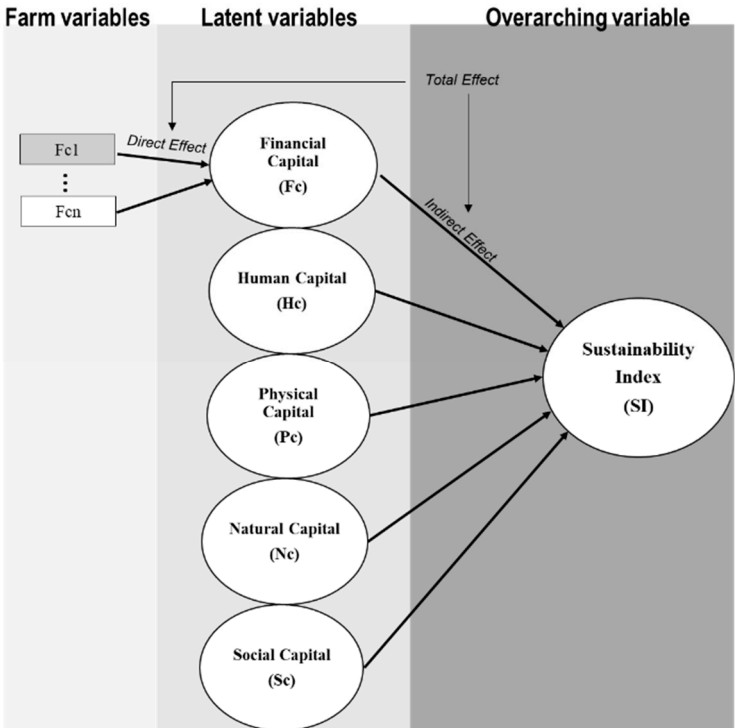

**Figure 2.** Path diagram of the confirmatory factor analysis model for the livelihood sustainability index (SI). The diagram is a graphic representation of the relationships among all the variables in our model. The observed farm variables are represented in a rectangular form, the capitals and the sustainability, represented in an elliptical form, are conceived as latent variables that cannot be measured directly, but can be inferred through a composed index of a set of observed farm variables. The relationships between variables are represented with straight arrows. SI: sustainability index, Fc: financial capital, Hc: human capital, Pc: physical capital, Nc: natural capital, Sc: social capital. Fc1.n are farm variables for financial capital.

Equation (1)

$$
\begin{aligned}
Fc &= Fc_1 + Fc_2 + \ldots + Fc_n + e_{Fc} \\
Hc &= Hc_1 + Hc_2 + \ldots + Hc_n + e_{Hc} \\
Pc &= Pc_1 + Pc_2 + \ldots + Pc_n + e_{Pc} \\
Nc &= Nc_1 + Nc_2 + \ldots + Nc_n + e_{Nc} \\
Sc &= Sc_1 + Sc_2 + \ldots + Sc_n + e_{Sc} \\
SI &= Fc + Hc + Pc + Nc + Sc + e_{SI}
\end{aligned}
\tag{1}
$$

where $SI$ = sustainability index, $Fc$ = financial capital, $Hc$ = human capital, $Pc$ = physical capital, $Nc$ = natural capital, $Sc$ = social capital. $Fc1..n$ = farm variables for financial capital, $Hc_1.n$ = farm variables for human capital, $Pc_1.n$ = farm variables for physical capital, $Nc_1..n$ = farm variables for natural capital, and $Sc_1.n$ = farm variables for social capital. $e$ is an error term.

The statistical analysis was conducted using R-project software (version 3.5.0). The statistical analysis package used was laavan through its function CFA [28]. The indices obtained for the five capitals were normalized to a range of 0-1 using the minimum-maximum normalization method. To analyze the validity of the model, goodness of fit for the indices was the root mean square error of approximation (RMSEA) below 0.08 and the comparative fit index (CFI) and the Tucker–Lewis index (TLI) were above 0.95.

### 2.4. Variability among Farms: Principal Component and Cluster Analyses

To analyze the relationships among the different capitals and provide a typology of farms grouped according to the similarities in regard to their SI values, a principal component analysis (PCA) was conducted. Cluster analysis grouped the farms into a small number of similar entities that were further characterized by their respective capitals.

### 2.5. Variability among Farm Landscapes: The Farm Landscape Indicator (Farmscape)

Farmscape has been defined as the composition, structure, and diversity of land covers within a farm [29]. Farmscaping is the result of farmers' strategy to diversify their farmland and thus their livelihoods. As a result, farmscape is closely linked to the set of production systems and the bundle of ecosystem services provided on a farm [30–32].

Farmscapes were described with a synthetic indicator that comprises the Shannon–Wiener index [33], the area intended for the different types of use such as fallow, pastureland, and natural areas (rivers, resting areas), and total farm size. The farmscape indicator is an objective measure of the different land-uses and landscape intensification. Additionally, it performs as an independent metric of the group of capitals, since the landscapes come from a continuous forest. To calculate it, we used the procedure proposed by Velasquez, E. et al. [34]. In the Amazonian sites studied by Lavelle, P. et al. [35], the landscape indicator decreased as biodiversity and soil-based ecosystem services decreased and the economic indicator increased. The ade4 library [36] in R-project software was used for these analyses. The farmscape indicator is described in Equation (2).

Equation (2)

$$Farmscape = Shannon\ index + \%\ fallow\ area + \%\ pastureland + \%\ natural\ areas + total\ farm\ size \quad (2)$$

where the Shannon–Wiener index [33] measures the entropy concept defined in Equation (3).

Equation (3)

$$H' = -\sum_{n=1}^{n}(p_i{}^*lnp_i) \quad (3)$$

where $p_i$ is the $i$th type of land-use inside the farm.

To analyze the variability among farmscapes, we compared the distribution of the SI and the farmscape indicator and their relationship. Furthermore, PCA and cluster analysis were carried out.

## 3. Results

### 3.1. General Household Characteristics

In the following, general summary statistics for the surveyed sample of households by farming system are presented. Of the 341 households interviewed in the study, 36% ranged in size from three to four people. In all households, on-farm activities were the main source of income, although in 61% of the cases, off-farm income was also an important income source. The average age of the household head was 50 and the highest level of education was limited to primary school (5 years of education). Only in 11% of the cases did females head the household, with males being the predominant decision-makers. As per the type of agricultural activities, households located in the rolling hills were mainly cattle ranchers and those located in mountain areas focused on subsistence cropping activities (Table 1).

Most of the households (93%) had secure tenure. The most common durable goods were cellphones (79%). Commonly, the houses had wooden floors and walls, and zinc roofs. Drinkable water was obtained from nearby streams. Most of the households had electric power and toilets connected to a septic tank. The principal source of fuel for cooking was firewood. According to the Latin American and Caribbean Food Security Scale (ELCSA) [37], close to 45% of the respondents suffered from mild food insecurity. Food scarcity usually occurred in the first four months of the year. The food products that were most frequently consumed were cereal, eggs, meat, and poultry. In contrast, the least frequently consumed products were fish and seafood, and vegetables.

<div align="center">**Table 1.** Characteristics of the household head.</div>

| | *n* | Males | | Females | | Average Age (Years) | Average Education (Years) | Main Occupation | | | |
|---|---|---|---|---|---|---|---|---|---|---|---|
| | | | | | | Mean | Mean | Cattle Rancher | | Farmer | |
| | | Freq. | % | Freq. | % | (S.D.) | (S.D.) | Freq. | % | Freq. | % |
| Livestock | 176 | 151 | 50.5 | 22 | 57.9 | 52.2 (13.3) | 5.53 (4.28) | 120 | 82.8 | 39 | 25.3 |
| Subsistence crops | 165 | 148 | 49.5 | 16 | 42.1 | 47.6 (13.7) | 4.49 (3.73) | 25 | 17.2 | 115 | 74.7 |
| Total | 341 | 299 | 100.0 | 38 | 100.0 | 50 (13.7) | 5.03 (4.05) | 145 | 100.0 | 154 | 100.0 |

<div align="center">Standard deviation in parentheses. In three observations, a household head was not identified.</div>

As per agricultural production activities, the households mentioned nine different livelihood activities. The proportions in which these were mentioned were different in the rolling hills vis-à-vis the mountain landscape (Figure 3). In the rolling hills, cattle ranching for milk production (80.11%) was the most common livelihood activity, followed by subsistence crops (51.13%) and aviculture (49.43%). In the mountains, subsistence crops were present on 86.06% of the farms, followed by cash crops (28.48%) and aviculture (25.45%).

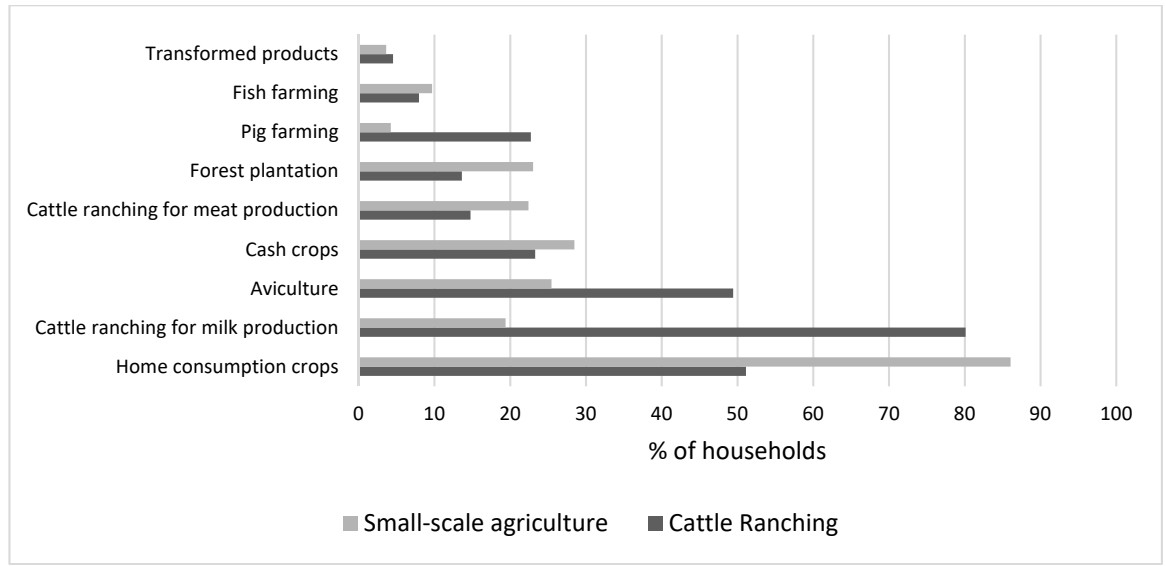

**Figure 3.** Productive activities reported on farms.

Other activities mentioned by more than 20% of the households in the rolling hills were pig farming, yet this was mentioned by only 4.24% of the households in the mountain landscape. Forest plantations and cattle ranching for meat production were also mentioned by more than 20% of the households in the mountain landscape.

On average, the farms in mountain areas were smaller (29.9 hectares, S.D. 25.04) than in the rolling hills (46.39 ha, S.D. 29.39).

The most frequent land-uses were pastures, fallow fields, forests, permanent crops, and, to a lesser extent, wetland. Other land-uses mentioned with low frequency were silvopastoral systems, temporary crops, agroforestry, and home gardens.

Finally, the most perceived common farm problems encountered were high prices of inputs, lack of technical assistance, degraded soils, and limited access to credit for coping with these problems. The best-known strategies that were mentioned the most to solve/overcome farm problems in the hill area were to establish silvopastoral systems (56%), reforest (51%), protect water sources (48%),

and use compost (47%). In the mountainous areas, farmers did not plan to do anything because of a lack of resources.

Most of the households surveyed were small-scale farms that relied to a certain extent on family members to carry out the farm's productive activities. Although in some cases labor is hired, farm work was predominantly conducted by family members. In the case of labor for cattle ranching, 57.39% of the households relied on family members for the application of vaccines or other medications, 55.11% for milking, and 42.05% for pruning management and weed control. Men conducted most of the farm labor, while women more often participated in milking. The percentage of households that reported women working in this activity was 16.9%. Hired labor was used only for specific seasonal activities such as pruning management, weed control (13.07%), and cropland preparation (18.18%).

Owning construction tools and machinery for productive activities is an essential indicator of the farm's technological level and household wealth. 77.27% of the households had agricultural and/or cattle ranching tools in the rolling hills, while this was much lower in the mountain zone (18.79%). In the mountain area, sheds for cows were found on only 29.55% of the farms and no construction or machinery was available on 78.18% of the farms.

The main productive activities were subsistence crops (68.03%) and cattle ranching for milk production (50.73%). Among the households that grow crops, lime stabilization was the most common practice (35.27%), followed by slash-and-burn (34.38%) and fallow (19.64%), while the least practiced activities were plowing (1.34%) and crop rotation (4.91%). In the rolling hills, cattle ranching was the predominant activity and bovines are the most common type of livestock, but horses, pigs, and poultry were also reared. The most common extension services mentioned were best practices for cattle ranching and crop management.

Regarding natural resources and the environment, nearly 70% of the households conserved forest patches to protect water sources, provide shade, and avoid erosion. However, only 18% of the households had reforested part of their farm to protect water catchments and nature. The collection of nontimber forest products was mainly for home use, firewood being most frequently used for cooking in the study area. Although households were conserving and reforesting in some cases, deforestation in the area still occurred for pasture, cash crops, and home consumption crop establishment.

### 3.2. Capitals and the Sustainability Index

Indices for the five capitals of the SLF and the SI were calculated using confirmation factor analysis. Given the significant differences between both land-use systems, the model was applied separately.

### 3.2.1. Selected Variables

The cattle ranching production system was characterized by 22 variables and small-scale family agriculture by 24 in total (Table 2). These variables were statistically significant in the model.

**Table 2.** Variables used to evaluate the five capitals of the cattle ranching and small-scale family agriculture systems.

| Capital | Cattle Ranching | Small-Scale Family Agriculture |
|---|---|---|
| Financial | Livestock unit<br>Milk production<br>Number of cows producing milk<br>Cattle ranching is the main occupation<br>Distance to town<br>Pasture areas | Family labor<br>Number of crops<br>Financial problems<br>Costs for agriculture and animal supplies<br>Use of herbicides<br>Extraction of firewood |
| Human | Maximum educational level<br>Pasture management<br>Food supply to household<br>Application of best production practices | Household Dietary Diversity Score<br>Good practices in crop management<br>Home consumption<br>Productive activities |

**Table 2.** *Cont.*

| Capital | Cattle Ranching | Small-Scale Family Agriculture |
|---|---|---|
| **Social** | Farmer association membership/participation<br>Education level of the household head<br>Marital status<br>Social acceptance<br>- | Household size<br><br>Training<br>Association<br>Maximum education level<br>Communication |
| **Physical** | Technological level of the home<br>Material goods<br>Total farm area<br>Technological level of the farm<br>- | Technological level of the home<br>Materiality index of housing<br>Total farm area<br>Electricity<br>Distance to town |
| **Natural** | Number of tree species extracted from the forest (timber forest products)<br>Reforestation<br>Number of areas related to natural resources<br>Number of species extracted/hunted (no timber forest products)<br>- | Number of tree species extracted from the forest (timber forest products)<br>Reforestation<br>Activities with natural resources<br><br>Extraction of firewood<br><br>Fallow areas |

### 3.2.2. Effect of Farm Characteristics on the Different Capitals

We obtained indices with high internal validity and the model had acceptable goodness of fit (see Supplementary Material Table S2). Standardized coefficients of the CFA indicate the relative importance of the variables in characterizing farms (Table 3).

Financial capital index: This index is positively influenced by the number of cattle, the density per ha, milk production, and pasture area in the cattle ranching system. This capital is also negatively affected by distance to town. In the small-scale family agriculture system, the most significant positive variable was the household workforce, followed by monthly expenses for agriculture and the number of crops. Financial problems mentioned frequently have a negative impact on this capital.

Human capital index: The cattle ranching system is positively influenced by best production practices and pasture management, whereas in the crop farming system, human capital is influenced by the diversity of crop management practices. Engagement in producing subsistence crops such as plantain and cassava improves food security and the diversity of self-consumption and productive activities.

Social capital index: In cattle ranching systems, social capital largely depends on the existence of producer organizations and social acceptance gained from the practice of cattle ranching. In the agricultural system, access to training and information through media are the variables that have the most impact on the calculation of capital. The household's general education level is essential, as well as household size and affiliation with any organization.

Physical capital index: Both productive systems depend on technological level, home assets, and total farm area. However, in the intensive cattle ranching system, technological level has a higher impact. In addition, a longer distance to populated centers means lower access to markets. Access to electric power reflects the presence of infrastructure developed by public institutions and governments in both systems.

Natural capital: In the cattle ranching system, households harvest timber, fruits, and animals from the forest. In comparison, in the agricultural system, households use firewood and timber, and have activities associated with natural resources (fishing, hunting). In both systems, natural capital is positively influenced when reforestation practices increase forest resources.

**Table 3.** Standardized coefficients of confirmatory factor analysis for sustainable livelihood, with all variables having significant variations among farms ($p < 0.05$).

| Capital | Cattle Ranching System | | Small-Scale Agriculture System | |
|---|---|---|---|---|
| Financial | Number of cows producing milk | 0.79 | Familiar workforce | 0.84 |
| | Livestock unit | 0.71 | Monthly expenses for agriculture (inputs) | 0.75 |
| | Milk production (cow/day/liter) | 0.61 | Number of crops | 0.59 |
| | Pastures Areas | 0.45 | Herbicide use | 0.43 |
| | Occupation of the household head (rancher) | 0.34 | Financial problems | −0.30 |
| | Distance to populated centers (Kilometers) | −0.56 | | |
| Human | Good productive practices | 0.86 | Productive activities | 0.95 |
| | Pasture management | 0.79 | Home use activities | 0.64 |
| | Maximum educational level | 0.19 | Household Dietary Diversity Score (HDDS) | 0.63 |
| | Food Supply | −0.21 | Agricultural practices applied to crops | 0.47 |
| Social | Social acceptance | 0.62 | The home received training | 0.87 |
| | Marital status | 0.27 | Membership of an association | 0.77 |
| | Livestock association | 0.18 | Home communication media (television, cellphone, computer, stereo) | 0.42 |
| | | | Household size | 0.28 |
| | | | Education of the household head | 0.25 |
| Physical | Total farm area | 0.54 | The technological level of home | 0.72 |
| | The technological level of home | 0.53 | Housing quality index | 0.62 |
| | The technological level of the farm | 0.40 | Electric power | 0.57 |
| | Housing quality index | 0.39 | Total farm area | 0.36 |
| | | | Distance to populated centers (Kilometers) | −0.28 |
| Natural | Non-timber forest products | 0.96 | Collection of firewood | 0.70 |
| | Timber-forest products | 0.66 | The farm has been reforested | 0.65 |
| | Different areas where it extracts natural resources | 0.63 | Timber-forest products | 0.56 |
| | The farm has been reforested | 0.59 | Number of activities with natural resources made by the household | 0.52 |
| | | | Percentage of resting areas | −0.31 |
| Sustainability Index | Financial capital | 0.83 | Human capital | 0.93 |
| | Human capital | 0.80 | Natural capital | 0.89 |
| | Physical capital | 0.80 | Social capital | 0.86 |
| | Natural capital | 0.71 | Physical capital | 0.57 |
| | Social capital | 0.60 | Financial capital | 0.49 |
| Number of observations: | | 164 | | 139 |
| Degrees of freedom: | | 185 | | 225 |

Note: a variable can belong to more than one capital since the analysis of the landscapes was done separately. For example, in the case of cattle ranching, the distance to the populated centers is a proxy for market access. In the case of small-scale agriculture, where the market is not crucial, the distance represents the existence and friction (state of the roads) of access roads.



A higher score from 0 to 1 represents greater contributions to sustainability by each capital. In cattle ranching systems, the regression coefficient obtained for financial capital (0.82) had the greatest effect on the sustainability index. This was followed by human capital (0.80), physical capital (0.80), and natural capital (0.70), and last was social capital (0.60). In agricultural systems, human capital had the greatest effect on the sustainability index (0.93), followed by natural capital (0.89), social capital (0.86), physical capital (0.56), and financial capital (0.48).

### 3.3. Variability among Farms

Farm variability within both farming systems and its five component capitals were analyzed with principal component analysis and typology of the farms according to their respective capitals. In both farming systems the five capitals vary together, which means that significant covariances exist among them. In the cattle ranching system, factor 1 of the PCA, which explained 69.6% of the total variance, separated farms on the horizontal axis, where those that had the highest values for all capitals are on the left end and those with the lowest values are on the right end (Figure 4). Factor 2 (21.7%) separated farms on the vertical axis, where those with high financial capital are on the lower end and those with high physical capital on the upper end. The cluster analysis recognized four well-individualized groups (Figure 4). A similar sketch (Figure 5) was obtained for the small-scale family agriculture system. The PCA shows a separation of farms with the highest index values between farms with high physical capital and farms with high financial capital. In fact, this system had five clusters, as opposed to cattle ranching with four (Figure 6).

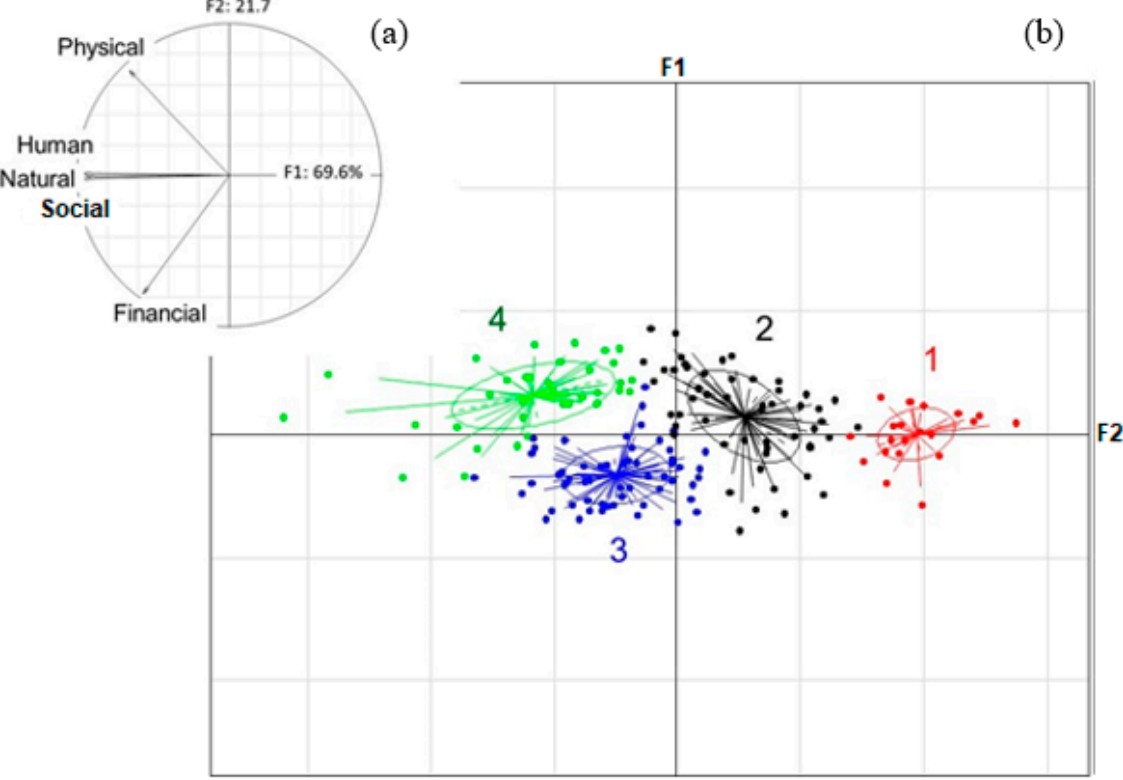

**Figure 4.** Projection of capital indices (**a**) and farms of the cattle ranching system distributed in four clusters (**b**) in the factorial plane F1F2 of a principal component analysis.

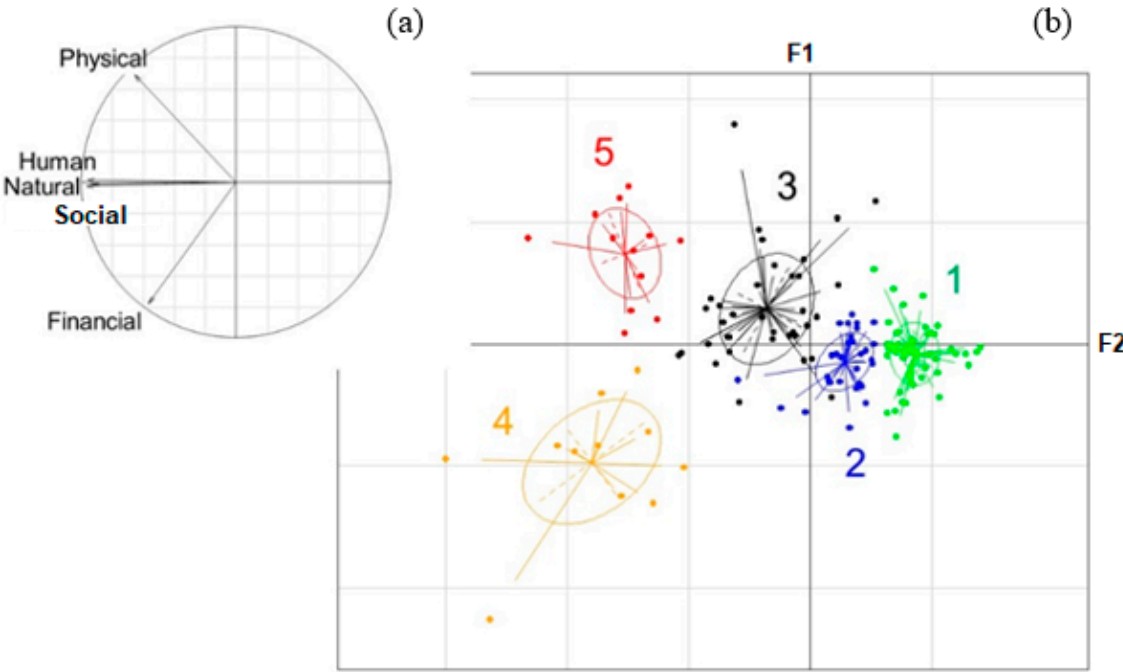

**Figure 5.** Projection of capital indices (**a**) and farms of the small-scale family agriculture system distributed in five clusters (**b**) in the factorial plane F1F2 of a principal component analysis.

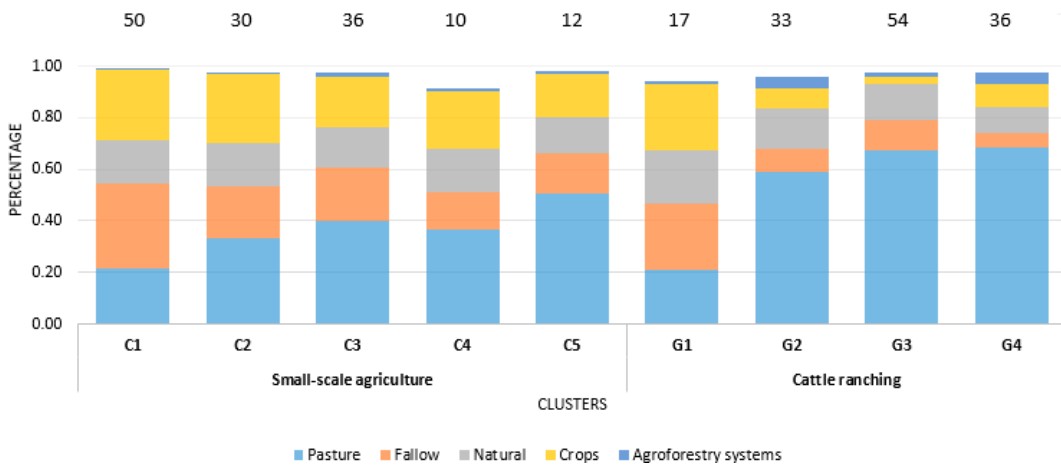

**Figure 6.** Land-uses by the different groups of farms. Small-scale family agriculture was characterized by more diversified land-use, while in cattle ranching, pastures are predominant. Above are the numbers of farms per cluster.

All capitals increased from cluster 1 to cluster 4 (Figure 7).

Although we observed two different clusters (G4 and G5) in the case of cattle ranching and one in the case of small-scale agriculture (C4), in which high values were presented in the five capitals, we wanted to contrast this measure of sustainability with the diversity and presence of large natural and regenerating areas. For this, an independent measure of the capitals was calculated. The landscape indicator collected the characteristics of the landscape in terms of overspecialization and conservation. Later, we evaluated the relationship of the mentioned metric with the results obtained from the cluster analysis.

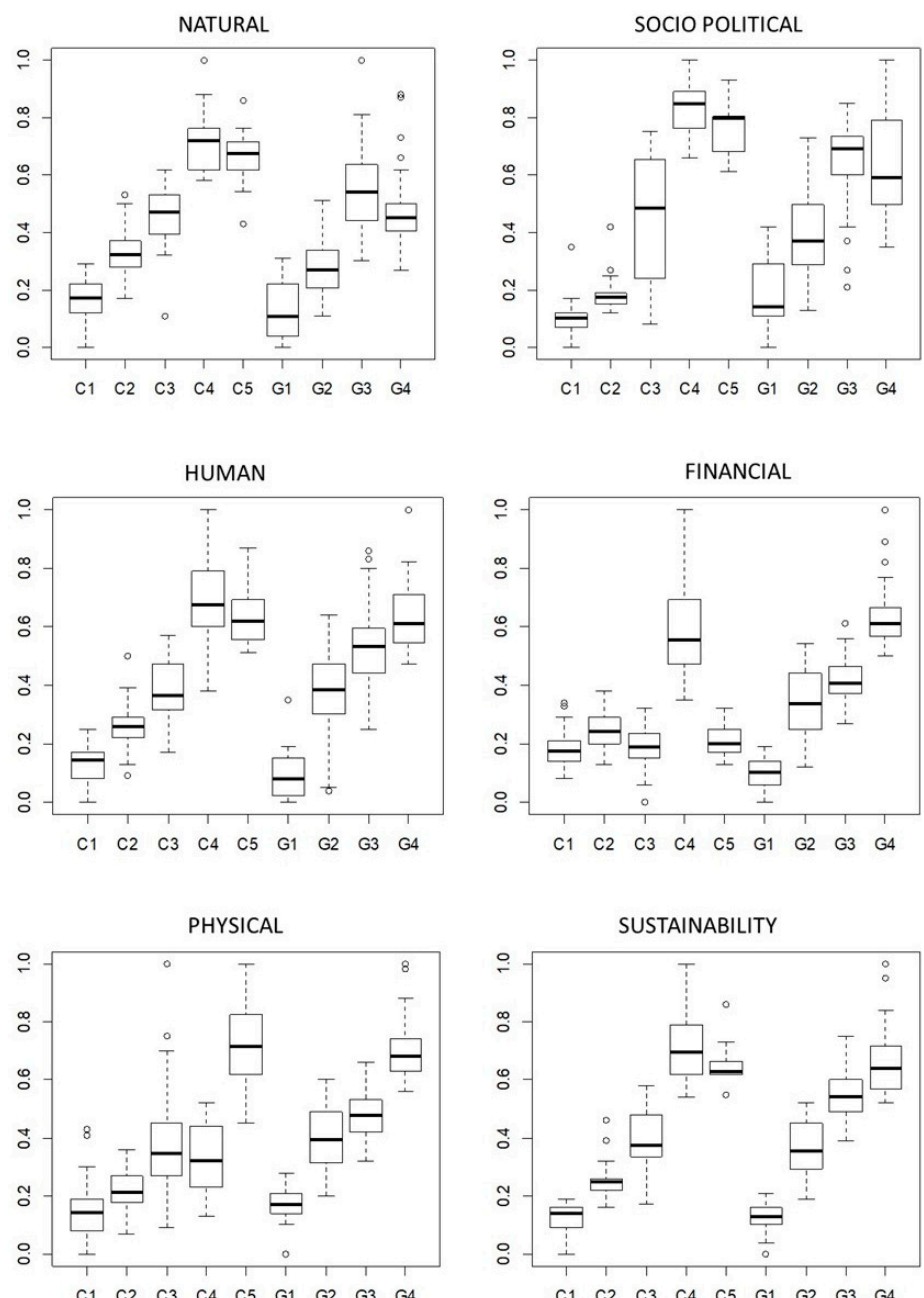

**Figure 7.** Variations in the five capital indices and sustainability index among clusters of the two systems. C: small-scale agricultural system, G: cattle ranching system. C1 to C5: small-scale family agricultural clusters; G1 to G4: cattle ranching clusters.

### 3.4. Variability among Farmscapes

We tested the hypothesis that the variations observed in capitals and sustainability indices were linked to farmscapes represented by the allocation of the total farm area to different land-uses and their diversity captured in the Shannon index. Five types of land-use were distinguished: crops, pastureland, silvopastoral systems, fallow, and natural areas.

With factor 1 (35.9% of variance explained), the analysis separated small farms mainly dedicated to agriculture and the most diverse composition for large farms with dominant pasture areas. Factor 2 (17.5%) separated farms according to the abundance of natural and regenerating areas. Factor 3 (14.3%, not shown) separated farms with agro-silvopastoral systems from the others. Figure 8a shows

that a large farm area was dominated by pastures, contrary to small farms, which had a diverse landscape and were dedicated mostly to agriculture.

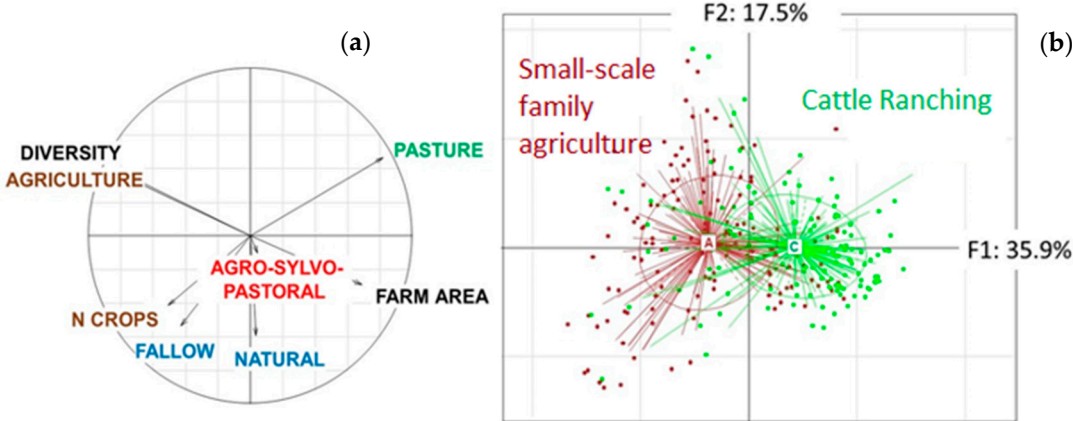

**Figure 8.** (**a**) Projection of farms from the two different areas on factorial plane F1F2 of a principal component analysis. (**b**) Projection of the cluster analysis: The separation between farms from the small-scale family agriculture (A) and cattle ranching (C) system was significant ($p < 0.01$, with 12% of variance explained).

Although the Monte Carlo test showed a significant difference between farmscapes of the hilly and mountain areas, there was considerable overlapping of farm projection points, which can be interpreted as large diversity inside each landscape.

The farmscape indicator built according to the procedure proposed by Velasquez, et al. [34] has maximum values where farmscapes are diverse and comprise large natural and regenerating areas. The farmscape indicator was maximum in clusters C1 and G1, with the lowest values of all capitals for clusters G4 and C5, and the highest values of all capitals in the two different areas (Figure 9a).

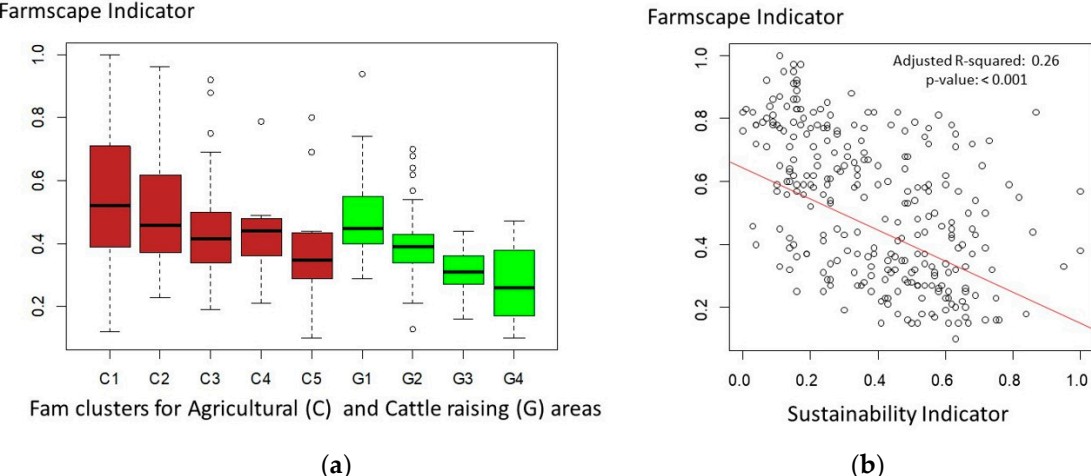

**Figure 9.** Variations of the farmscape indicator according to farm clusters (**a**) and correlation of the farmscape indicator with the sustainability index (**b**).

The sustainability index was inversely related to that of farmscape: the most transformed landscapes without natural areas exhibited the highest SI values. All capitals were inversely correlated with the farmscape indicator. This was especially true for natural capital.

## 4. Discussion

### 4.1. Which Indicators (Positive or Negative) Determine the Level of Sustainability in Farm Covariation of Capitals?

Clear positive and negative indicators were identified for all capitals. However, in the cattle ranching system, sustainability seemed to depend on financial, physical, and human capital, more so than natural and social capital. Optimal values of sustainability occurred on large farms located close to populated centers, where large numbers of cattle units produced high amounts of milk. Good production practices, adequate pasture management, and access to forest timber or nontimber products positively contribute to sustainability. On the other hand, insufficient food supply due to low subsistence farming is a severe threat to sustainability in these systems [38].

The small-scale family agricultural system exhibited a different profile, with greater importance of the family workforce and training, technological level, and crop diversity. Financial restrictions due to the elevated cost of external agrochemical inputs and fluctuating prices for cash crops are significant threats. The high input costs and their threat of financial bankruptcy to agricultural systems have been widely demonstrated in many places, including Colombia [39,40].

Farms were classified into four clusters in the cattle ranching system and five clusters in the small-scale family agriculture system. The clusters were grouped from the least sustainable, with the lowest values for each capital, to the most sustainable, with maximum values for all capitals. For the cattle ranching system, we observed a trade-off between physical and financial capital increase and natural capital improvement. This result was expected due to the high rates of deforestation associated with livestock production and this shows the need for the implementation of sustainable land-use systems. According to our model, the most sustainable farms were the ones that had fewer natural areas and more grasslands (see Figure 6), a consequence of the importance given in the model to financial capital in the cattle ranching system.

This finding seems similar to some studies demonstrating that economic and human development in Amazonia has always taken place at the expense of natural capital. [41], for example, show a bright "bust" in human development indices of Brazilian Amazonian municipalities some 30 to 40 years after the initial "boom." Another study conducted on farms from Colombian and Brazilian Amazonia also shows contrasting variations of the economics and the environmental index [35].

### 4.2. How Does the Farmscape Affect Sustainability?

At our study sites, farmscapes exhibited significant differences between the two farming systems examined and within them. However, despite a significant global difference between cattle ranching and small-scale agriculture systems, a considerable degree of overlapping was observed. Overall, the SI was higher as land-use diversity was higher and large areas were dedicated to natural vegetation, restoration, or fallow. A significant inverse correlation was observed between the SI and the farmscape indicator for both landscapes. This showed a trade-off between landscape degradation and human development consisting of transforming natural areas into progressively degraded production systems while physical and financial capitals increase transitorily.

### 4.3. How Could We Increase Sustainability?

The problems faced by farmers are different depending on the geographic and socioeconomic context they deal with, which is reflected in their production systems implemented. At the studied sites, financial issues were the main concern in the small-scale family agriculture system vis-à-vis low productivity and food insecurity in the cattle ranching system. Nevertheless, we can argue that excessive specialization is a common factor that decreases sustainability in both of them, for which the adoption of mixed production systems could be a solution.

Improving sustainability in the cattle ranching farming system requires an increase in production efficiency to have a healthier herd and increase in milk production in the same area [42,43]. Silvopastoral

systems are clear options since they increase carrying capacity and milk production and quality with changes operating in feeding cows with a more diversified set of herbaceous and ligneous plants and using protein banks [44,45]. However, increased cattle densities also mean higher excretion rates and risks of eutrophication and contamination of local water resources, in addition to increased zoonotic pressure. Finally, cattle ranching systems can hardly be termed ecologically sustainable, given the large areas required to produce fodder and the extremely high carbon emissions per kg of animal product produced.

The agricultural farms of the mountain area face critical financial problems, especially related to the purchase of external inputs in a context of uncertain value paid for production [39]. This problem can be solved with the production on the farm and the application of organic fertilizer. Furthermore, the positive effect of crop numbers indicated in our results shows the necessity of diversification of crops and landscape composition and structure.

Human capital, as defined in our study, will benefit from an increase in the use of good practices on farms [42]. On the other hand, the deficiency of food supply may be offset by integrating food corridors in silvopastoral systems [2].

Improving physical capital assumes a rise in the technological level of the farm. Obtaining more cowsheds with a roof and obtaining machinery for productive activities are two examples. Technological improvement of the farm is related to an increase in production efficiency, which is a general target of sustainable land-use practices.

The improvement of natural capital is conditioned by on-farm conservation and reforestation activities as well as the production of timber and nontimber forest products [40,44]. Some specific silvopastoral systems have a clear focus on this type of production, stressing the fact that production of timber trees will likely secure farmers' pension when the trees are harvested, some 30 years later [40,44,46,47]. The natural capital considered in our study comprises the goods and services that people take from natural areas. Surprisingly, this capital was inversely correlated with our farmscape index that has its highest values when the farmscape is diverse and has large areas of natural or restored areas.

Social capital, mainly determined by affiliation with associations and the value of social networks, is directly related to the presence of government and nongovernment entities as well as the self-organization of producers in the area. The traditional design and implementation of silvopastoral systems might be an opportunity to reinforce these links [46,47].

Human capital in a small-scale farming system could be improved by increasing and making better capacity building for farmers to learn about best production practices that will increase land-use and livelihood diversification.

Physical capital shows the importance of the technological level of the home that is directly connected with social capital in terms of connection to associations as well as with human capital with access to information. On the other hand, a longer distance to populated centers has a negative impact on sustainability, but this variable could be improved through public policies oriented to improving ways and promoting public transportation facilities.

### 4.4. Questioning the Conceptual Framework

The evaluation of sustainability depends on how the interaction of the capitals that compose it is interpreted [48]. In the literature, there are several interpretations of the reciprocal substitution between the economic component (physical and financial capital) and natural capital. One of them refers to weak sustainability in which sacrificing natural capital can be defended by comparable gains in other capitals. Physical and financial capitals can replace the natural capital, since the total amount of stock is not declining. So, even if natural capital deteriorates until no recovery, it is still sustainable [48–50]. By contrast, a strong sustainability definition establishes that natural, physical, and financial capitals are not substitutes, but complementary [48,49,51]. In this definition, a healthy environment is necessary

to achieve development, since natural capital makes a unique contribution to welfare. In this definition, rapid economic growth accompanied by natural resource depletion is not sustainable [49,50].

The SLF fails in the weak sustainability definition, even though it allowed a well-structured and precise evaluation of farm households to provide an assessment of their sustainability index score and quantification of the different capitals. As it was applied, the SLF was more heavily weighted toward socioeconomic sustainability than an integrated socioecological sustainability measure, because among SI components, two capitals, financial and physical, represent the accumulation of goods and values, representing the economic aspects of sustainability. The inverse relationship between farming intensification and the loss of natural areas is in contrast with the strong definition of sustainability. We consider this as a weakness of the SLF, which leads to the formulation of public policies aimed at increasing sustainability being more oriented to economic dynamics than to environmental protection. In the long run, sustainability should be possible only when positive synergies occur, when the environmental component is in a good state as well as the economic and social components [49,50]. Then, despite the attributes of this conceptual framework, it is necessary to look critically at its use and perhaps consider a conceptual framework in which the environmental component has a more significant predominance. This means it is necessary to evaluate the Colombia Amazonia into a strong sustainability definition which also adheres to the constant capital rule, but without compromising natural capital [49].

Costanza, R., et al. [52] and Reid, W.V., et al. [53] emphasized the importance of a correct evaluation of ecosystem services in the face of commercial and marketable services. Additionally, they highlighted that, despite the importance of ecosystem services for human well-being, they are still little considered in policy decisions. To make this point, a different conceptual framework that provides higher weight to natural capital can be used. For example, Lavelle, P. et al. [35] showed a significant relationship between production systems, landscapes, and ecosystem services. They proposed a sustainability indicator, which is the sum of social, economic, biodiversity, and ecosystem services indicators. This indicator can also be applied to evaluate the environmental impact of agricultural activities. A conceptual framework such as this would allow giving greater importance to the use and conservation of natural resources, following the general beliefs of studied farmers themselves.

Finally, future researches should include possible sources of inequality in access to livelihood resources that do not allow achieving sustainability as defined in Scoones, I. [12]. Additionally, a single time frame sample can ignore important changes that drive drastic transformation [54]. Then, coming studies may be oriented to overcome this limitation by developing an analysis that captures the changes in the livelihood strategies, socioeconomic conditions and households' preferences in time [13]. Thus, forthcoming research should study the dynamic of the livelihood strategies trough livelihood pathways [54,55].

**Supplementary Materials:** The following are available online at http://www.mdpi.com/2077-0472/10/12/588/s1, Table S1. Household-level survey structure; Figure S1. Methodological process for the Sustainability index estimation; Table S2. Summary of the model fit information used to calculate the SLF indices.

**Author Contributions:** Conceptualization, L.P.M., P.L., M.R.J., A.C.-N.; methodology, L.P.M., P.L., J.C.S.; software K.C., P.L., J.C.S.; validation P.L.; formal analysis, L.P.M., P.L.; investigation M.A.R., A.S.; data curation K.C.; writing—original draft preparation L.P.M., P.L.; writing—review and editing, W.F., A.C.-N., M.V.-C.; supervision P.L., M.R.J., A.C.-N.; project leader M.Q., A.C.-N.; research design, M.Q., A.C.-N.; fund acquisition M.Q., A.C.-N. All authors have read and agreed to the published version of the manuscript.

**Funding:** This research contributes to the Project Sustainable Amazonian Landscapes (14_III_057_A_Latin America_Sustainable Development Options) and to the Project Sustainable Land Use Systems (18_III_106_COL_A_Sustainable productive strategies). These projects are part of the International Climate Initiative (IKI). The Federal Ministry for the Environment, Nature Conservation and Nuclear Safety (BMU) supports this initiative on the basis of a decision adopted by the German Bundestag. This study is framed in the CGIAR Research Program on Water Land Ecosystems (WLE). This article is part of Lisset Pérez's Ph.D. dissertation with the University of Copenhagen, Denmark.

**Acknowledgments:** We are grateful to Fabio Alexander Castro, Erwan Sachet, and Harold Achicanoy for their contributions to the manuscript. Finally, we express our gratitude to the local families from the municipalities of Caquetá (Colombia) for their time and cooperation to collect the data and accomplish the project objectives.

**Conflicts of Interest:** The authors declare no conflict of interest.

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
