# Peer review of "Farmscape Composition and Livelihood Sustainability in Deforested Landscapes of Colombian Amazonia"

_agriculture, doi:10.3390/agriculture10120588_

Round 1

Reviewer 1 Report

Reviewers comments:  Farmscape composition and livelihood sustainability in deforested landscapes of Colombian Amazonia

I find the paper well-written. It is clear and concise and efficiently presented. The paper makes important contributions to our empirical knowledge. The methods are sound, the analysis is sound and the interpretations insightful. 

Here, I  provide some general and specific comments for the authors and editor to consider.

First, I would like to see a few sentences in the introduction highlighting the limitations of the SLF. The paper adequately justifies why and how it employs SLF. However, for completeness, and potential implications in the discussion section, SLF’s strengths and weaknesses are important for understanding the study’s policy implications.

Specifically, I am thinking of the limitations of a single time-frame sample. Scoones (2009) reinforces that single time-frame analyses can fail to disclose potential for drastic transformations. This might be especially important in an Amazon farmscape. Livelihood strategies can be short-term, employed as a coping mechanism against an unanticipated event or a long term, well thought out incremental process to achieve bigger goals. These strategies can change over time as socio-economic conditions and households’ preferences evolve (Chambers and Conway, 1992). In other words, livelihood strategies transit through time and create livelihood pathways (De Haan and Zoomers, 2005; Scoones, 2009).

Scoones, I., 2009. Livelihoods perspectives and rural development. The Journal of Peasant Studies, 36(1), pp.171-196. De Haan, L. and Zoomers, A., 2005. Exploring the frontier of livelihoods research. Development and change, 36(1), pp.27-47.

Chambers, R. and Conway, G.R., 1991. Sustainable rural livelihoods: Practical concepts for the 21st century. 1991. Institute of Development Studies: Brighton.

 Second, Are there any reasons that the 2014 census does not capture all the 2016 farmers? Is this region relatively stable in terms of in and out migration. The information on the sample is perhaps to scant. A footnote or other means to reinforce the case that the sample is representative. What is the total population of cattle ranching and the proportion of the landscape? What is the total population of the small-scale family farms in the census and proportion of the landscape?

Third, are different variables making up Fc and Hc and Pc etc, all weighted equally?

Fourth, line 201, most household had land tenure. Does this mean ‘secure tenure’? For example, title with outright ownership. Every farm has some type of tenure:  private, common, public, open access, community, village, etc.

Fifth, I wonder if ‘affected’ is the best word to describe ‘association’. On lines, 277, 279, 283, 284 and 301, using affected seems to imply causality. Isn’t this just correlation?

Sixth, are the authors able to go beyond just suggesting what interventions and given the rich data set and their analysis suggest the how? For example, line 107-109: we can argue that excessive specialization is a common factor that decreases sustainability in both of them, for which the adoption of mixed production systems could be a solution. What do you recommend given your knowledge of the landscape and farmscape and households? How would you target different household clusters to incentivise mixed production systems? There are several example in the discussion section.

I hope these comments are useful.

Author Response

Dear Editor and reviewers,

We are pleased to submit the revised version of our manuscript entitled “Farmscape composition and livelihood sustainability in deforested landscapes of Colombian Amazonia” for consideration for publication in your journal.

We are grateful for the time and effort taken in assessing our manuscript, as well as for the opportunity to revise it. We appreciate the views of both Reviewers that our analysis offers important value and insight regarding the understanding of sustainability, and fully take on board their suggestions to improve our discussion on sustainability. We argue that our paper contributes to understanding sustainability in the Colombian Amazonia, which is poorly understood, but also believe our study contributes broadly to the discussion on sustainability by helping separate the contributions and importance of different variables in several landscapes.

We have addressed all the comments by amending the paper, to the best of our ability. Suggestions to move text/tables to the main text and future comments are welcome.

In this revision, we attach a changes-tracked version of the manuscript, as well as a point-by-point response to all of the comments from the both Reviewers.

We look forward to hearing from you in due course, and hope that this revised manuscript meets the requirements of Agriculture.

Sincerely yours,

Lisset Pérez Marulanda (corresponding author) on behalf of all authors.

Lisset Pérez Marulanda, Msc.

PhD. Candidate University of Copenhagen, Denmark

Research senior associate at the Alliance Bioversity-CIAT, Colombia

Reviewer 2 Report

The paper uses the Sustainable Livelihoods Framework to characterise sustainability amongst agricultural practices in department of Caquetá, Colombia. Statistical methods are used to develop cluster analysis to further explore the relationship between and within the main  agricultural livelihood typologies in the region. This work is potentially an important example of how a framework might be used to guide decision making processes with sustainable agriculture. There is some interesting and valuable data here.

Problem: The fundamental issue here would seem to be that the SI is built from the capitals as opposed to be compared to an independent SI. As a result financially successful but environmentally destructive farms are described as sustainable.  The high values for Financial and physical capital do not mean that the system is sustainable - just that the economic part of sustainability is met - it is the balance between the capitals. 

Suggestion:

1) Identify an independent metric of sustainability - possibly through land use change using remote sensing?  

2) Focus on the clusters that have high levels of all capitals - these seem to have the best potential sustainability? Possibly map them?

As a reviewer who mainly works with statisticians as opposes to being one I would defer to a statistician on this – but the current written material would seem to support the interpretation above

Abstract

This might include a bit more detail of what is meant by an “equation system”

“From the results, the capitals that most contributed to a higher SI score were financial and physical”. This sentence needs some explanation – The sustainability comes from the balance between these capitals ( as mentioned briefly in the Introduction). This appears to be saying the system is sustainable because it predominantly produced money and infra-structure? I f this was not what was intended then please edit

Introduction

The authors do talk about sustainability being a balance between capitals but could expand upon what that means in this region in the introduction. A little more depth on sustainable and non sustainable practices in this region.

L81 – This suggests that farmers will be gathering and using this data? Would it not be better used by decision marks and local government?

L88 – Authors say that they want to contrast this method with a standard method? What method is that? Please outline – SLF is a very standard method itself? Is this a local used approach?

Would expect the map to be in the introduction. The map should include some detail of the land cover or use in the study area to give context. The current map gives a rough idea of location only.

Method ( needs to be checked by statistician)

L107 -  You mention the proposition of surveys conducted in each agricultural group but not the number of households from which the sample was taken – what % does your sample represent of the census households in the area.

L140 – missing reference

Figure 2 – This needs much more explanation – what are latent variables?  What do direct variables mean? Need to make this clear. The reader can probably work it out but it needs to be explained. Surely the arrows should run from the variables to sustainability – at the moment they run from sustainability?

 -Unable to comment on the choice of equations

 - There is no discussion about weighting of the direct and indirect variables. It is fine to say you used equal weighting but there needs to be some discussion about this

- Surely the capitals can be built  through some common variables across the 2 systems so that they are more comparable.  Income  to input ration for Finance for example – and land use change with time for natural capital – it is not clear that the 2 systems are really comparable using the current approach of highly different components for the capitals of each agricultural system

Results

L193      “Of the 341 households interviewed in the study, 36% ranged in size from three to four people”. What is the significance of this and why is this the opening line of the results?

L203 – onwards. There is a lot of numbers here that might be better represented in a table of results – hard to read a mass of statistics and hard to make comparisons or links. Possibly select the important ones and tabulate – can the rest go in an appendix?

Table 2 – is the title correct? Seems to cover both groups

“A higher score from 0 to 1 represents greater contributions to sustainability by each capital. In 302 cattle-ranching systems, the regression coefficient obtained for financial capital (0.82) had the 303 greatest effect on the sustainability index. This was followed by human capital (0.80), physical 304 capital (0.80), and natural capital (0.70), and last was social capital (0.60). In agricultural systems, 305 human capital had the greatest effect on the sustainability index (0.93), followed by natural capital 306 (0.89), social capital (0.86), physical capital (0.56), and financial capital (0.48).”

The above is unclear  - The capitals might be used to build a SI – with balance between them over time being the important factor.   Here there seems to be an independent measure of sustainability against which it is possible to determine the relationship to the individual capitals. Is this right – if so that does not come across – what is the independent measure of sustainability? If not then it is unclear what the above paragraph means?

Not sure you should use same variable in more than one capital? This need justification

The Clustering work which shows that there are systems within both agricultural groups that have high levels across all capitals are surely the most sustainable? This would seem to counter the point you make about physical and financial capitals being the dominant in sustainability

Discussion

“Clear positive and negative indicators were identified for all capitals. However, in the 64 cattle-ranching system, sustainability seems to depend on financial, physical, and human capital, more so than natural and social capital.”

“According to our model, the most sustainable farms were the ones that had fewer natural areas and more grasslands, a consequence of the importance given in the model to financial capital in the cattle ranching system.”

The above would seem to say that cattle farms that are financially successful and have infra-structure are necessarily sustainable?  Surely it is the relationship between this financial success and natural and social capital that is the measure of sustainability? Why not use your clustering results to identify the systems that appear to have a better balance?

Author Response

(The authors gave the same response as above.)

Round 2

Reviewer 2 Report

Most of the issues raised have been addressed. I would ask for minor modification before publication please

1) "The independent metric of the group of capitals is the landscape indicator, and it is a terminal indicator of change because the landscapes come from a continuous forest."

Could you please make the use of this as your independent variable clearer in the abstract and method

2) It is not normal practice to include a single indicator in more than one capital - however as you explain a why you have chosen to do this clearly then others can decide.

3) Your conclusions around sustainability being higher where Finance Physical and Human capital are higher would benefit from greater explaination. The trade off you are suggesting does make sense in a mature system and I would ask that you frame this within the concepts of Weak and Strong sustainability then I think this will enhance your discussion. You are proposing a Weak model of sustainability over a strong one. 

Author Response

Dear Editor and reviewer,

We are pleased to submit, in a second round, the revised version of our manuscript entitled “Farmscape composition and livelihood sustainability in deforested landscapes of Colombian Amazonia”.

We are grateful for the time and effort taken in assessing our manuscript several times. We are pleased to publish in your journal. In this version, we respond to the suggested minor changes.

Please, find attached a changes-tracked version of the manuscript, as well as a point-by-point response to the comments from the Reviewer.

We are attentive to your response and future comments are welcome.

Sincerely yours,

Lisset Pérez Marulanda (corresponding author) on behalf of all authors.

Lisset Pérez Marulanda, Msc.

PhD. Candidate University of Copenhagen, Denmark

Research senior associate at the Alliance Bioversity-CIAT, Colombia
